# The Association Between Maternal Urinary Iodine Concentration and Neonatal Anthropometry

**DOI:** 10.3390/nu17101624

**Published:** 2025-05-09

**Authors:** Simon Shenhav, Leah Tsur Shenhav, Dov Gefel, Shani R. Rosen, Amit Shenhav, Rachel Shapin, Eyal Y. Anteby, Yaniv S. Ovadia

**Affiliations:** 1Obstetrics and Gynecology Division, Barzilai University Medical Center, Ashkelon 7830604, Israeleyala@bmc.gov.il (E.Y.A.); yaniv.ovadia@mail.huji.ac.il (Y.S.O.); 2Faculty of Health Sciences, Ben-Gurion University of Negev, Beersheba 8410501, Israel; shapinra@post.bgu.ac.il; 3Faculty of Medicine, The Hebrew University of Jerusalem, Jerusalem 9190500, Israel; leahtsur@gmail.com; 4Medical Corps, Israel Defense Forces, Ramat Gan 52625, Israel; 5School of Nutritional Science, Institute of Biochemistry, Food Science and Nutrition, Robert H. Smith Faculty of Agriculture, Food and Environment, The Hebrew University of Jerusalem, Rehovot 7610001, Israel; dubigefel@gmail.com (D.G.); shani.rosen1@mail.huji.ac.il (S.R.R.); 6Ruth and Bruce Rappaport Faculty of Medicine, Technion, Israel Institute of Technology, Haifa 3525433, Israel

**Keywords:** iodine, urinary iodine concentration, birth weight, neonatal anthropometry

## Abstract

**Background/Objectives**: Iodine deficiency disorders remain a global public health concern, as acknowledged by the World Health Organization (WHO). Adequate maternal iodine intake during pregnancy is essential for normal fetal development, yet the relationship between maternal iodine status and fetal growth remains controversial. Urinary Iodine Concentration (UIC) is a commonly used marker for assessing iodine status. This study evaluates the association between maternal UIC and neonatal anthropometric parameters. **Methods**: This prospective single-center cohort study included 202 pregnant women without known or reported thyroid disease, recruited between 2018 and 2021. Maternal iodine status was assessed by UIC from spot urine samples collected at the time of recruitment. Correlations were analyzed between maternal UIC and neonatal anthropometric measures, including birth weight (g), length (cm), and head circumference (cm). Analyses stratified by fetal sex were also performed. **Results**: No statistically significant association was found between UIC and neonatal anthropometric measures. Analysis of these correlations, stratified by fetal sex, did not reveal any statistically significant associations either. **Conclusions**: Maternal UIC showed no association with neonatal anthropometric outcomes in this study, regardless of fetal sex. Further research is needed to investigate the additional effects of maternal iodine status in healthy, euthyroid pregnant women on neonatal outcomes.

## 1. Introduction

Iodine deficiency (ID) disorders in pregnant women are a global health problem, both in developing and developed countries, according to the World Health Organization (WHO). An estimated 35% of the world’s population has insufficient iodine intake and is therefore at risk of ID disorders. ID is recognized as one of the primary preventable causes of brain damage in newborns worldwide [1,2].

Iodine is an intrinsic component of thyroid hormones, which are essential for normal fetal development. During the first 18–20 weeks of gestation, the fetus is dependent on the placental transfer of maternal thyroid hormones; therefore, their production is increased [3,4,5]. Inadequate iodine intake during this critical period may disrupt hormone synthesis. Severe ID increases the risk of pregnancy complications, including intrauterine death, low birth weight (LBW), preterm labor, small head circumference, and various levels of neurological deficits [6].

Iodine status can be assessed by Urinary Iodine Concentration (UIC), an economical and readily available measure. Since most of the iodine absorbed by the body is excreted in the urine, UIC is regarded as a sensitive marker of current iodine consumption that can reflect recent changes in iodine status. Although UIC may vary daily or even within the same day, these fluctuations tend to even out when testing a broad population [7].

According to the World Health Organization, normal UIC levels in pregnant women range from 150 to 249 μg/L. The recommended iodine intake for a non-pregnant adult is 150 μg per day; however, during pregnancy, the demand increases to 250 μg per day [2]. In Israel, a national iodine survey was conducted for the first time in 2016, based on UIC tests of elementary school-aged children (*n* = 1023) and pregnant women (*n* = 1074). The survey revealed mild to moderate ID among a representative sample of Israeli pregnant women, with 85% of the participants exhibiting UIC falling below the WHO recommended range, and a median UIC of 61 μg/L [8]. Although it was recommended more than ten years ago, there is currently no legal requirement in Israel to enrich salt with iodine.

It is known that there is a link between hypothyroidism in pregnant women (clinical or subclinical) and low birth weight [9]. Nonetheless, the potential consequences of maternal iodine status in euthyroid women on fetal growth remain uncertain. In a systematic review by Farebrother et al., it was unclear whether prenatal iodine repletion increases infant growth among pregnant women with ID [10]. On the other hand, a prospective study by Rydbeck et al. involving 1617 pregnant women from Bangladesh demonstrated a positive relationship between UIC and the weight, length, and head circumference of male newborns, whose mothers had a UIC of up to 1 mg/L [11]. Overall, the available evidence regarding the relationship between maternal iodine status in euthyroid women and neonatal anthropometric outcomes remains conflicting [12,13].

In this study, we aimed to describe the correlation between maternal iodine status and the anthropometry of their newborns, including weight, length, and head circumference, and to further investigate the effect of fetal sex on each of these parameters.

## 2. Materials and Methods

This prospective cohort study is part of a series of studies conducted in the Obstetrics and Gynecology Department at Barzilai University Medical Center, Ashkelon, Israel (BUMCA). These studies examined the iodine status of pregnant women in the city of Ashkelon and its surrounding areas (Ashkelon sub-district), as well as its effects on various variables in both the women and their newborns. This study was approved by an ethics committee and followed the tenets of the Declaration of Helsinki.

### 2.1. Participants, Settings, and Design

A total of 202 pregnant women were recruited from the Obstetrics and Gynecology Department at BUMCA between May 2018 and March 2021. Recruitment took place in both outpatient clinics and inpatient settings, including the high-risk pregnancy unit. All participants provided written informed consent. Participants who had any reported or known underlying thyroid disease, including clinical hypothyroidism and/or positive thyroid antibodies (TgAb levels > 40 IU/mL and TPOAb levels > 35 IU/mL), were excluded from this study [14]. Participants with high levels of TgAb were also excluded because TgAb may disrupt the analysis of Tg levels in the blood [14]. In addition, volunteers who had undergone a fine needle aspiration (FNA) test within 15 days prior to this study, received iodine-containing contrast material within the three months prior to this study, were taking iodine-containing medications (including Amiodarone, Phenytoin, and/or lithium-containing drugs), had a multiple pregnancy, or had missing data were also excluded.

### 2.2. Study Protocol

For each participant, a non-fasting venous blood sample (up to 10 mL) was drawn within 24 h of recruitment to test thyroid function, including TSH, free thyroxine (FT4), thyroglobulin (Tg), antibodies to the peroxidase enzyme (TPOAb), and Tg antibodies (TgAb). The samples were immediately centrifuged, and the serum was separated and stored at −20 °C until analysis. Analysis was performed by the clinical biochemistry laboratory at BUMCA using electrochemiluminescence immunoassay on the Modular Analytics E601 analyzer (Cobas, Roche Diagnostics GmbH, Mannheim, Germany).

Maternal iodine status was assessed using UIC. A spot urine sample was collected into an iodine-free cup within 4 h of recruitment. Each sample was centrifuged, and the urine specimen was transferred into vacutainer tubes containing stabilizers (chlorhexidine, ethylparaben, and sodium propionate) without dipstick testing, which could otherwise contaminate the iodine assay. The specimens were then refrigerated and delivered to the National Biomonitoring Laboratory of the Israeli Ministry of Health for UIC analysis. UIC was determined using the inductively coupled plasma mass spectrometry method (ICP-MS) according to the US CDC ICP-MS method [15], using an Agilent 7800 ICP-MS Instrument (Santa Clara, CA, USA), equipped with an Integrated Sample Introducing System and High Matrix Introducing mode. A median UIC value lower than 150 μg/L was considered indicative of insufficient iodine intake, in accordance with WHO recommendations [1].

Neonatal birth data—including gestational age, sex, Apgar score, birth weight (g), length (cm), and head circumference (cm)—were obtained from the participants’ medical records at BUMCA and linked to the corresponding maternal spot UIC result. Weight and length percentiles were calculated according to the Israeli birth population references: Dollberg et al. for weight percentiles [16] and Davidson et al. for length percentiles [17], both adjusted for gestational age and sex.

### 2.3. Statistical Analysis

Statistical analyses were performed using SPSS version 25.0 (IBM Corporation, Armonk, NY, USA) and JMP Pro software, version 16 (SAS institute, Cary, NC, USA). The strength and direction of associations between UIC levels and other quantitative variables were assessed using Spearman’s rank correlation coefficient, a non-parametric test. Comparisons of UIC levels between two independent groups were conducted using the non-parametric Mann–Whitney U test. To simultaneously assess the effects of two independent variables on a quantitative outcome, a multivariable linear regression model was applied. Associations between categorical variables were evaluated using Fisher’s exact test. Non-parametric tests were selected due to the non-normal distribution of the main study variable, UIC. For maternal UIC values across the total study population and at the extremes of birth weight percentiles, crude individual UIC measurements were log-transformed prior to analysis to normalize their distribution. Geometric mean and standard deviation (SD) were then calculated from the back-transformed data. All statistical tests were two-tailed, with *p* ≤ 0.05 considered statistically significant.

## 3. Results

### 3.1. Study Population

Of the 202 pregnant volunteers, 172 participants and their newborns were included in this study. Thirty of the volunteers met the exclusion criteria, which were as follows: underlying thyroid diseases (*n* = 4), multiple pregnancy (*n* = 7), CT with contrast material (*n* = 1), positive thyroid antibodies (*n* = 10), and missing data (*n* = 8).

The maternal population had an average age of 30.88 ± 5.24 years. A total of 70.7% of the participants were born in Israel. The average BMI (kg/m^2^) before pregnancy was 24.25 ± 4.75. Additionally, 7.6% of the participants (*n* = 13) were diagnosed with gestational diabetes (GDM), and 15.3% (*n* = 26) reported smoking. These characteristics are summarized in Table 1.

The average gestational age of the newborns was 38.48 ± 2.16 weeks, with 14% (*n* = 24) classified as preterm labor (before the 37th week of pregnancy). A total of 94 newborns (54.7%) were male, and 78 (45.3%) were female. The average birth weight was 3087.97 ± 606.15 g, the average length was 50.42 ± 2.9 cm, and the average head circumference was 34.19 ± 1.88 cm. After adjusting the birth weights to the Dollberg et al. [16] percentile, 8.2% of the newborns were placed above the 90th percentile, and 8.8% were placed below the 10th percentile. Birth lengths were adjusted to the Davidson et al. [17] percentile, with 42.3% of the newborns placed above the 90th percentile and 1.9% below the 10th percentile. These neonatal characteristics are presented in Table 2.

### 3.2. UIC Values and Their Relation to the Anthropometric Measures

The median UIC and the geometric mean ± SD were 59.4 μg/L and 60.8 ± 2.1 μg/L, respectively, with a wide range of 3.9–1389.7 μg/L. A UIC below the WHO-defined threshold of 150 μg/L was observed in 88.4% of the study participants.

Spearman’s correlation coefficient revealed no statistically significant relationship between UIC and the examined anthropometric measures. For weight, the correlation coefficient between UIC and birth weight (g) was −0.109, and the coefficient between UIC and Dollberg et al. percentiles was −0.124 (Figure 1A). For length, the correlation coefficient between UIC and length (cm) was −0.142, and the coefficient between UIC and Davidson et al. percentiles was −0.147 (Figure 1B). The correlation coefficient between UIC and head circumference (cm) was −0.004 (Figure 1C).

An assessment of UIC values relative to the extremes of birth weight percentiles, both above 90% and below 10%, was also conducted. No statistically significant differences were found in the UIC values across these groups, with *p* = 0.232 for newborns above the 90th percentile and *p* = 0.264 for newborns below the 10th percentile, as presented in Table 3.

### 3.3. Effect of Stratification by Newborn Sex on the Results

Following a similar study conducted in Bangladesh [11], which identified sex-specific differences in the effect of UIC on anthropometric measures, UIC values were analyzed separately by newborn sex. No statistically significant difference in UIC levels was observed between the groups (*p* = 0.418), as shown in Figure 2. When Spearman’s correlation coefficients were calculated separately by sex, no statistically significant associations were found between UIC and the examined anthropometric parameters, as shown in Figure 3A–C. Among females, the correlation coefficients for weight percentiles, length percentiles, and head circumference were −0.102, −0.023, and −0.043, respectively. Among males, the correlation coefficients for weight percentiles, length percentiles, and head circumference were −0.139, −0.211, and −0.039, respectively.

## 4. Discussion

This prospective cohort study examined the relationships between UIC and neonatal growth parameters, including birth weight, length, and head circumference. Our findings did not support the hypothesis of an existing association between maternal iodine levels and neonatal anthropometry, nor were significant differences found after stratification by fetal sex.

This study demonstrated a significant ID among pregnant women in the research area (Ashkelon sub-district, Israel). A preliminary study [18], in which maternal iodine status was assessed using dietary questionnaires, found that only 41% of pregnant women consumed an adequate amount of iodine. Our findings were even more pronounced, with 88.4% of participants having UIC values below the WHO recommendations. The median UIC was 59.4 μg/L, compared to the recommended range of 150–249 μg/L [1]. This finding supports the previously mentioned study, which attributed ID to both reduced intake and the consumption of desalinated water, which contains negligible iodine levels.

Previous studies have investigated the relationship between maternal iodine status and various pregnancy outcomes. A cohort study conducted by Snart et al. found no association between UIC and adverse pregnancy outcomes, including intrauterine fetal death, preterm birth rates, intrauterine growth restriction, or differences in head circumference [19]. This study included 6971 pregnant women from the UK, with urine samples collected between 26 and 28 weeks of gestation. Unlike our study, however, it reported that newborns of iodine-deficient pregnant women had, on average, a birth weight that was 41 g lower than those born to women with normal iodine levels. Although both the UK and our study populations were iodine deficient (median UIC: 76.2 vs. 59.4 μg/L, respectively), possibly indicating the absence of ethnicity- or region-related confounding, differences in UIC–birthweight associations may be explained by variations in population characteristics, timing of urine sampling, or both. In the UK study, urine samples were collected at the beginning of the third trimester, and crude birth weight was reported [19], whereas in our study, most participants were sampled in the later stages of the third trimester (Table 1), and neonatal birth weight percentiles were used as the primary outcome. It is also possible that the narrower gestational window (26–28 weeks) in the UK study may have fallen outside a critical developmental period relevant to fetal growth, or that iodine status alone may not account for the population-adjusted birthweight scale applied in the current study [16].

An additional study conducted by Gargari et al. [20] measured UIC in 884 pregnant women living in an iodine-rich region of Iran. This study found that only 5.2% (*n* = 46) of the participants were iodine-deficient, compared to 88.4% in our study population. In contrast to Snart et al., Gargari et al. identified a significantly higher risk of preterm labor in iodine-deficient pregnant women compared to those with normal UIC, with an odds ratio of 3.29 (95% CI: 1.51–7.1). In our study, the preterm birth rate was 14%, which is approximately twice the rate observed in the general population in Israel [21]. However, this elevated rate may have been influenced by selection bias, as some participants were recruited while hospitalized in a high-risk pregnancy unit.

Despite the significant percentage of preterm births in the study by Gargari et al., no significant association was found between maternal UIC and birth weight, length, or head circumference. Similarly, a meta-analysis by Nazeri et al. [22], which included 11 studies investigating the potential impact of UIC and thyroid hormone levels on neonatal anthropometric parameters, found no association between UIC levels and birth weight, head circumference, or length.

The largest and most recent meta-analysis on the subject, conducted by Greenwood et al. in 2023, included data from 23 cohorts comprising a total of 42,269 participants [12]. The study assessed the association between maternal iodine status and various birth outcomes, concluding that a higher maternal UIC (>150 μg/L) was associated with a lower risk of small for gestational age (SGA) infants, with a relative risk of 0.85 (95% CI: 0.75–0.96, *p* = 0.01). While the link between maternal UIC and the risk of SGA infants was evident, its association with birth weight, length, and head circumference was modest and did not reach statistical significance.

Nevertheless, other studies have found associations between maternal UIC and fetal anthropometric measures. A prospective study conducted by Rydbeck et al. in Bangladesh found that a maternal UIC of up to about 1.0 g/mL was positively associated with birth weight, length, and head circumference only in male newborns, suggesting a sex-specific effect of maternal iodine status on fetal growth parameters [11]. Possible explanations for such sex-specific differences include the greater sensitivity of male fetuses to variations in maternal iodine levels, potentially due to differences in thyroid hormone metabolism or placental function between the sexes [11]. In our study, however, stratification by fetal sex did not reveal any statistically significant differences in UIC values between the sexes, nor in the associations between UIC and neonatal anthropometric measures. These findings suggest that fetal sex may not have a significant impact on the relationship between maternal UIC and fetal growth, though additional studies are needed to confirm this across different populations.

An alternative approach to assessing maternal iodine status involves the use of the iodine-to-creatinine (I/Cr) ratio. This method accounts for urinary dilution, possibly making it a more stable indicator of iodine status [23]. Greenwood et al. [12] demonstrated that while UIC levels were inversely associated with the risk of SGA infants, the I/Cr ratio did not show a statistically significant relationship. However, a nonlinear association was observed between I/Cr and birth weight, with levels up to 200 μg/g correlating with increased birth weight. This suggests that I/Cr may be more sensitive in detecting subtle variations in iodine status that affect fetal growth. Nevertheless, UIC remains the widely used and straightforward measure for population-level iodine status assessment, as recommended by the American Thyroid Association (ATA) [24].

Although our study did not reveal an association between UIC and neonatal anthropometrics, future research may further investigate the intricate relationship between maternal iodine status, placental weight, and fetal growth and anthropometrics. Bienertová-Vašků et al. identified a significant inverse relationship between estimated dietary iodine intake and placental weight, suggesting a potential role of iodine in placental development and function [25]. Neven et al. also reported variability in placental iodine concentration, indicating that placental iodine storage may reflect long-term maternal iodine status and potentially impact fetal growth [26]. Such investigations are particularly warranted in pregnant populations with overt hypo- or hyperthyroidism or severe ID, as the current study population was euthyroid (Section 2.1) and classified as having mild to moderate ID (severe ID is defined as a median UIC < 50 μg/L, according to ATA guidelines) [27].

This study has several limitations. First, there was variability in the timing of urine sample collection, as specimens were obtained at recruitment rather than at a standardized gestational week. While most participants enrolled during the third trimester, a smaller proportion joined during the first and second trimesters. Given that maternal iodine status may fluctuate during different stages of pregnancy due to thyroid gland development in the fetus [28], this variability may have influenced the observed associations. Second, UIC values were not adjusted for creatinine, which may lead to misinterpretation of iodine status due to hydration-related variations. We assumed that participants had similar hydration status, supported by evidence that dehydration tends to increase UIC values; therefore, any resulting bias would likely have led to an underestimation of ID prevalence [8]. Thus, the high prevalence of low UIC values observed in our study population (Figure 1 and Figure 2) supports the robustness of this finding. Third, the sample size in our study was significantly smaller than in similar studies, such as the study by Rydbeck et al. [11] and meta-analyses [12,19] involving thousands of women and newborns. However, the high prevalence of ID in our study enabled a more thorough exploration of the potential impact of maternal iodine status on neonatal anthropometry, yielding similar results and providing insight into sex-specific effects.

## 5. Conclusions

In conclusion, our study contributes to the existing literature on the relationship between maternal iodine status and neonatal anthropometry, suggesting that maternal UIC is not associated with neonatal anthropometric parameters, even after stratification by fetal sex. Additionally, our findings highlight the high prevalence of ID in the studied region, emphasizing the possible need for targeted public health interventions. Further investigation is warranted to elucidate the potential effects of maternal iodine status on both fetal and placental development, as well as on various birth outcomes.

## Figures and Tables

**Figure 1 nutrients-17-01624-f001:**
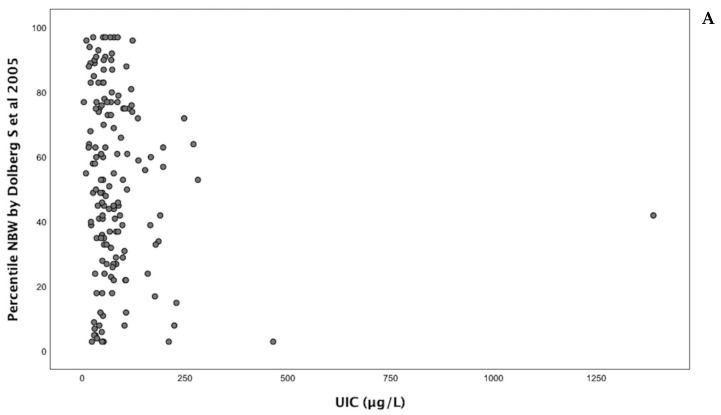
Correlations between UIC and weight percentile (**A**), length percentile (**B**), and head circumference (**C**) of newborns. UIC = Urinary Iodine Concentration, NBW = Neonatal Birth Weight, NBL = Neonatal Birth Length [16,17].

**Figure 2 nutrients-17-01624-f002:**
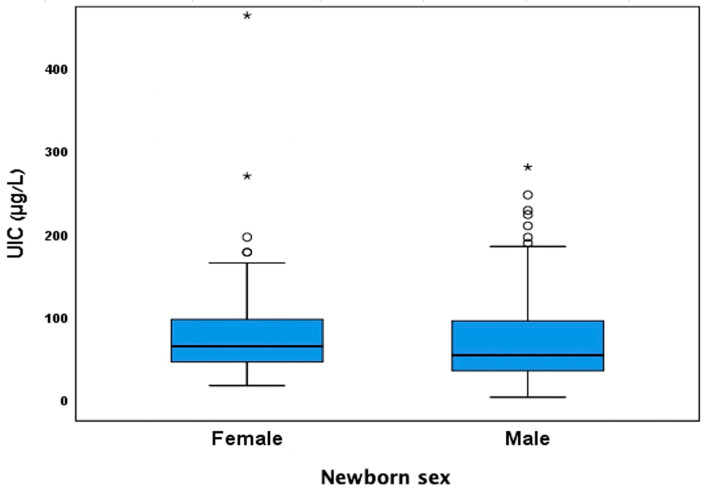
UIC values categorized by newborn sex. UIC = Urinary Iodine Concentration. Small circles represent mild outliers; asterisks represent extreme outliers.

**Figure 3 nutrients-17-01624-f003:**
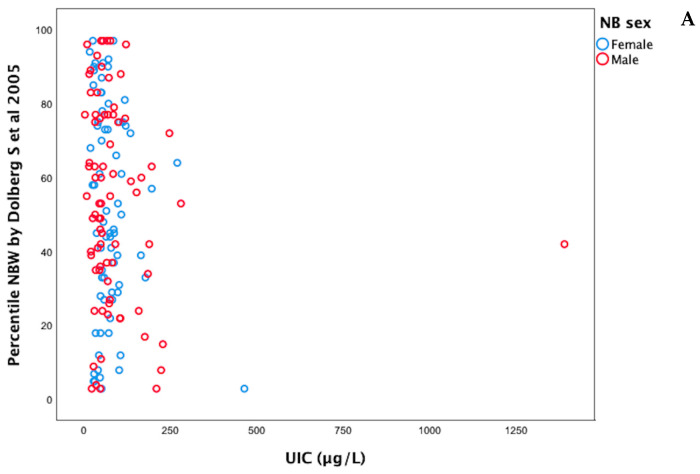
Correlations between UIC and weight percentile (**A**), length percentile (**B**), and head circumference (**C**) of newborns by sex. UIC = Urinary Iodine Concentration, NBW = Neonatal Birth Weight, NBL = Neonatal Birth Length [16,17].

**Table 1 nutrients-17-01624-t001:** Descriptive characteristics of the maternal population.

	N	
Maternal Age (y)	171	30.88 ± 5.24
Gestational week at recruitment (weeks)	162	32.45 ± 6.21
Gravidity	171	3 (1–10)
Parity	172	2 (1–10)
Country of birth	167	
Israel	118 (70.7%)	
Other	49 (29.3%)	
Current smoking habits	170	
Not smoking	144 (84.7%)	
Smoking	26 (15.3%)	
BMI (kg/m^2^)		
BMI before pregnancy	125	24.25 ± 4.75
BMI (in recruitment)	171	28.62 ± 5.10
BMI change	126	4.44 ± 2.29
GDM diagnosis in current pregnancy	171	
No	158 (92.4%)	
Yes	13 (7.6%)	

Data are presented as mean ± SD, median (range), or N (%). BMI = body mass index, GDM = gestational diabetes mellitus, SD = standard deviation.

**Table 2 nutrients-17-01624-t002:** Descriptive characteristics of the newborn population.

	N	Mean ± SD
Birth week	172	38.48 ± 2.16
Birth week < 37	24 (14%)	
Birth week > 37	148 (86%)	
Newborn sex	172	
Male	94 (54.7%)	
Female	78 (45.3%)	
Apgar score	170	
Apgar 5 min < 7	0 (0%)	
Apgar 5 min ≥ 7	170 (100%)	
Newborn birth weight (g)	172	3087.97 ± 606.15
Percentile NBW by Dollberg et al. (2005) [16]	171	52.34 ± 27.27
>90th percentile	14 (8.2%)	
<10th percentile	15 (8.8%)	
Length (cm)	156	50.42 ± 2.90
Percentile NBL by Davidson et al. (2008) [17]	156	72.77 ± 26.06
>90th percentile	66 (42.3%)	
<10th percentile	3 (1.9%)	
Head Circumference (cm)	157	34.19 ± 1.88

NBW = newborn weight, NBL = newborn length, SD = standard deviation.

**Table 3 nutrients-17-01624-t003:** Comparison of UIC values in the extremes of birth weight percentile.

UIC (μg/L)	BW < 10th Percentile by Dollberg		BW > 90th Percentile by Dollberg	
	No	Yes	No	Yes
N	137	14	138	13
Mean ^BT^ ± SD ^BT^	62.48 ± 2.08	59.90 ± 2.52	64.12 ± 2.12	48.74 ± 1.91
Median	61.93	44.58	60.02	56.13

^BT^ back transformed data. UIC = Urinary Iodine Concentration, BW = birth weight, SD = standard deviation.

## Data Availability

The data presented in this study are available on reasonable request from the corresponding author due to ethical reasons.

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
