# Peer review of "The Association Between Maternal Urinary Iodine Concentration and Neonatal Anthropometry"

_nutrients, 2025, doi:10.3390/nu17101624_

Round 1
Reviewer 1 Report
Comments and Suggestions for Authors
The Association between Maternal Urinary Iodine Concentration and Neonatal Anthropometry
Reviewer’s comments
The authors have incurred much time cost conducting this research and writing the manuscript. Iodine nutrition during the neonatal period is essential to brain development and infant growth. I have some suggested edits and comment to help improve this manuscript.
- Materials and methods; study protocols, Line 93: Based on your phrase “for each female participant”, were some of the participants males? Please edit to reflect if some of the participants were males.
- Materials and methods; Study protocols, Lines 112-114: It is not clear what the authors mean by “retrospectively”. Please provide more details about the timing of the neonatal data collection and which specific pregnancy provided the neonatal data.
- Materials and methods; Statistical analysis, Line 117: Creatinine correction is used to adjust for hydration status when UIC is analyzed. Did the authors adjust for hydration status? If you have creatinine data, you can include it in the regression model to correct for hydration status. If not, please include text in the limitations section at page 10 of this manuscript: It was assumed that participants had similar hydrate status, so hydration status was not corrected during the data analysis involving UIC.
- Materials and methods; Statistical analysis, Line 126: Please edit p-value≤0.05 to read p≤0.05.
- Table 1, page 3-4: The authors don’t need to report both the mean and median where only one of them is correctly represented.
- Results: All acronyms should be explained in a footnote under each table of results or figure.
- Results, Table 3, page 7: Generally, UIC data is heavily skewed, that is why in your study the standard deviation is larger than the mean or median. To report the correct mean value and standard deviation or median, the UIC data needs to be log-transformed to correct for skewness (normality). Usually after the log-transformation, regression models could be applied. The mean log-transformed values can then be back-transformed to obtain the correct mean values.
- Page 9, line 201: The mean should be calculated from the log-transformed UIC data.
- Figures 1 and 4: With the exception of the head circumference, the correlation scatter plots should be the correlation between the actual birthweight or body length with UIC values. Doing correlation after applying clinical cut-offs or regrouped data will blunt the sensitivity of the analysis.
- Please correct Figure 2 title. Replace “divided” with “categorized”
Author Response
Thank you very much for your comprehensive and constructive feedback, and for taking the time to review our manuscript. We have carefully addressed each point raised and have made corresponding revisions to our manuscript. These changes are indicated in the revised manuscript using the Track Changes function for easy identification.
Point-by-point response to comments and suggestions:
Comment 1: Materials and methods; study protocols, Line 93: Based on your phrase “for each female participant”, were some of the participants males? Please edit to reflect if some of the participants were males.
Response 1: Thank you for pointing that out. Indeed, all of the study’s participants were pregnant women, and data were collected for their newborns as well. Therefore, the phrase “female participant” was unnecessary and has been removed throughout the manuscript for better clarity (Lines 92, 101, and 142).
Comment 2: Materials and methods; Study protocols, Lines 112-114: It is not clear what the authors mean by “retrospectively”. Please provide more details about the timing of the neonatal data collection and which specific pregnancy provided the neonatal data.
Response 2: We have clarified the text to explain that neonatal data were collected from the newborns’ medical records and were linked to the specific pregnancy from which the maternal spot urine sample was obtained. The text has been revised accordingly. (page 3, lines 119–122).
​​Comment 3: Materials and methods; Statistical analysis, Line 117: Creatinine correction is used to adjust for hydration status when UIC is analyzed. Did the authors adjust for hydration status? If you have creatinine data, you can include it in the regression model to correct for hydration status. If not, please include text in the limitations section at page 10 of this manuscript: It was assumed that participants had similar hydrate status, so hydration status was not corrected during the data analysis involving UIC.
Response 3: Thank you for this comment. As suggested, we have added this limitation to the manuscript (page 10, lines 307–313).
Comment 4: Materials and methods, Statisical analysis, Line 126: Please edit p-value≤0.05 to read p≤0.05
Response 4: Corrected (page 3, line 138).
Comment 5: Table 1, page 3-4: The authors don’t need to report both the mean and median where only one of them is correctly represented.
Response 5: Thank you for the comment. Both Table 1 (pages 3-4) and Table 2 (pages 4-5) have been revised to report either mean ± SD, median (range) or n (%), depending on the distribution of each variable.
Comment 6: Results: All acronyms should be explained in a footnote under each table of results or figure.
Response 6: All tables and figures have been to ensure that all acronyms are properly defined. Missing footnotes were added where necessary (lines 152-153, 165, 189–190, 204).
Comment 7: Results, Table 3, page 7: Generally, UIC data is heavily skewed, that is why in your study the standard deviation is larger than the mean or median. To report the correct mean value and standard deviation or median, the UIC data needs to be log-transformed to correct for skewness (normality). Usually after the log-transformation, regression models could be applied. The mean log-transformed values can then be back-transformed to obtain the correct mean values.
Response 7: Thank you for this very constructive comment. We have reanalyzed relevant measures of central tendency and dispersion by applying log-transformation to the UIC data. The geometric mean and corresponding standard deviation were calculated based on the back-transformed results. These changes are reflected in the revised statistical analysis section (page 3, lines 134–137), and the revised values are now presented in Table 3 (page 7).
Comment 8: Page 9, line 201: The mean should be calculated from the log-transformed UIC data.
Response 8: The mean value has been revised and is now calculated from the log-transformed UIC data. (Page 5, line 167).
Comment 9: Figures 1 and 4: With the exception of the head circumference, the correlation scatter plots should be the correlation between the actual birthweight or body length with UIC values. Doing correlation after applying clinical cut-offs or regrouped data will blunt the sensitivity of the analysis.
Response 9: Thank you for the comment. As our newborn population varied by gestational age at birth, we believe that correlating UIC to crude birthweight and length (unadjusted for gestational age and sex) is beyond the scope of this manuscript. However, we have double-checked Figures 1 and 4 and confirmed that they do not include clinical cut-offs or regrouped data that might blunt the sensitivity of the analysis regarding birthweight and length.
Comment 10: Please correct Figure 2 title. Replace “divided” with “categorized”.
Response 10: We have corrected the title of Figure 2 as suggested (line 203).
Reviewer 2 Report
Comments and Suggestions for Authors
The manuscript “The Association between Maternal Urinary Iodine Concentration and Neonatal Anthropometry” provides novel information on the relationship between Iodine concentration in urine and neonatal anthropometry. Unfortunately, their correlations were not significant, but I personally believe that these basic results can contribute to application to clinical research in the future. The authors need to provide additional explanation and discussion of the results obtained.
- We know that maternal iodine intake during pregnancy is important for normal fetal development. However, the correlation between them showed no significance in this study. What are some possible reasons for the results in the manuscript? Does iodine deficiency vary by race, region, and so on?
- Why do the authors think there is a difference in anthropometry between the sex of neonate by the maternal iodine intake during pregnancy?
- Iodine is necessary for thyroid hormone synthesis. However, based on the results of this study, maternal thyroid hormone during pregnancy may not be essential for normal neonatal development. The authors should add further discussion about this point.
The English could be improved to more clearly express the research.
Author Response
Thank you very much for taking the time to review our manuscript and for your comments and observations regarding our study. We have carefully considered each of your comments and made the corresponding revisions throughout the manuscript. These changes are indicated using the Track Changes function in the resubmitted manuscript.
In addition, we have thoroughly reviewed and improved the English throughout the manuscript to ensure greater clarity.
Point-by-point response to comments and suggestions:
Comment 1: We know that maternal iodine intake during pregnancy is important for normal fetal development. However, the correlation between them showed no significance in this study. What are some possible reasons for the results in the manuscript? Does iodine deficiency vary by race, region, and so on?
Response 1: Thank you for this insightful comment. We have expanded the Discussion section to address possible explanations. (page 9, lines 231–242).
Comment 2: Why do the authors think there is a difference in anthropometry between the sex of neonate by the maternal iodine intake during pregnancy?
Response 2: We have expanded the Discussion section to address possible explanations for sex-specific differences in the relationship between maternal iodine status and neonatal anthropometry, as suggested by previous studies (pages 9-10, lines 266–278).
Comment 3: Iodine is necessary for thyroid hormone synthesis. However, based on the results of this study, maternal thyroid hormone during pregnancy may not be essential for normal neonatal development. The authors should add further discussion about this point.
Response 3: Thank you for highlighting this important point. We have added a relevant discussion regarding this issue (page 10, lines 296-300).
Round 2
Reviewer 1 Report
Comments and Suggestions for Authors
Figure 2. Since you have enough space, you might want to spell out “NB” to “New born”.
Reviewer 2 Report
Comments and Suggestions for Authors
I think that the manuscript was improved.